# Assessment of multiple herbicide protection seed treatments for seed-based restoration of native perennial bunchgrasses and sagebrush across multiple sites and years

Owen W. Baughman[1]*, Magdalena Eshleman[2], Jessica Griffen[1], Roxanne Rios[3‡], Chad Boyd[3‡], Olga A. Kildisheva[4], Andrew Olsen[5‡], Matthew Cahill[4‡], Jay D. Kerby[1,6‡], Corinna Riginos[2]

1 The Nature Conservancy of Oregon, Burns, Oregon, United States of America, 2 The Nature Conservancy of Wyoming, Lander, Wyoming, United States of America, 3 Agricultural Research Service, United States Department of Agriculture, Burns, Oregon, United States of America, 4 The Nature Conservancy, Bend, Oregon, United States of America, 5 Intermountain West Joint Venture, Missoula, Montana, United States of America, 6 Prineville, Oregon, United States of America

☯ These authors contributed equally to this work.
‡ These authors also contributed equally to this work.
* owen.baughman@tnc.org

**Data Availability Statement:** The data underlying the results presented in the study are available in

## Abstract

The invasion of exotic, annual plant species is a leading contributor to ecological degradation in drylands globally, and the use of pre-emergent herbicide to control these species is common. Pre-emergent herbicides pose challenges for seed-based restoration due to toxicity to the seeds of desired species. Herbicide protection (HP) technologies pose a potential solution by using activated carbon seed treatments to protect desirable seeds from herbicide exposure. In the sagebrush steppe ecosystem of North America, we used an adaptive small plot design over three planting years to test for effects on seeding outcomes (seedling density and size) of large and small multi-seed HP pellets, several single-seed HP coatings, and carbon banding treatments at geographically dispersed sites for several perennial bunchgrasses and the keystone perennial shrub, Wyoming big sagebrush. We also compared different methods of seed delivery and litter pre-seeding management. Seeding success was low overall, especially for sagebrush, and it was clear that other, often less predictable barriers to establishment than herbicide exposure, such as inadequate spring moisture, were strong drivers of seeding outcomes. Despite this, HP treatments were associated with higher seedling density than bare seed in multiple instances, most notably for grasses. The large HP pellet occasionally outperformed the small HP pellet, and several HP coatings performed similarly to the small pellet. Surprisingly, we did not see consistent negative effects of pre-emergent herbicide on unprotected bare seed. We conclude that HP seed treatments show some promise to improve seeding success in the presence of herbicide, but that consistent success will require further improvements to HP treatments as well as integration with other innovations and approaches.

the Dryad repository (https://doi.org/10.5061/
dryad.vq83bk3xp).

**Funding:** This work was based on research funded
by the United States Department of Agriculture –
Agricultural Research Service (https://www.ars.
usa.gov) under agreement #58-2070-7-003 (OB,
JK, OK), The Nature Conservancy (www.nature.
org; OB, JK, OK, ME, JG, CR), and the United
States Department of the Interior – Fish and
Wildlife Service (https://www.fws.gov) under grant
#F21AC01957-00 (OB, OK, AO). CB and RR
received no specific funding for this work. The
funders had no role in study design, data collection
and analysis, decision to publish, or preparation of
the manuscript.

**Competing interests:** The authors have declared
that no competing interests exist.

## Introduction

Invasive annual grasses are a major driver of ecosystem degradation in dryland regions of
North America [1,2]. The sagebrush shrub-steppe of the western United States is one of the
largest semi-arid, cold desert biomes in the world and once covered approximately 60.7 mil-
lion hectares [3]. This system is currently undergoing a large-scale conversion to dominance
by invasive annual grasses [4], which now occupy 23 million hectares of formerly sagebrush
and perennial grass-dominated communities [5]. Invasive annual grasses like *Bromus tectorum*
(cheatgrass; downy brome) alter fine fuel characteristics leading to more frequent and severe
wildfires [6,7]. This annual grass-fire cycle causes the loss of native plant cover and threatens
sagebrush-dependent species and rural economies [8–11]. Following large-scale invasion,
native plant communities typically cannot recover without active restoration [12].

Efforts to restore these systems to prevent further spread of invasive annual grasses are
extensive. For example, the United States Bureau of Land Management spends an average of
$21 M per year rehabilitating burned areas [13]. However, efforts to re-establish native plant
communities on highly invaded sites often fall short of restoration targets [14]. This is due in
part to the ability of invasive annual grasses to competitively suppress native plants [15,16],
particularly at the seedling stage [17,18].

Pre-emergent herbicides, like imazapic (BASF, Ludwigshafen, Germany), can be effective
tools in reducing invasive annual grass abundance [19,20] and enabling native plants to re-
establish in the absence of strong invasive competition. Once established, perennial vegetation
aids in community resistance to exotic annual grass dominance and can reduce the spread of
the annual grass-fire cycle [21,22]. However, herbicide application also negatively impacts
seeded species [23]; as a result, restoration seedings in this region are typically carried out one
or more years after treatment with pre-emergent herbicide, which may result in reinvasion
and restoration failure [24] and also requires multiple site entries.

Herbicide protection methodologies may offer an opportunity to simultaneously apply pre-
emergent herbicide and seed desired species in a single-entry approach, reducing invasive
annual grass competition on target seedlings for at least the crucial first growing season [25–
28]. Herbicide protection uses activated carbon, which can capture the applied herbicide and
create a safe site for desired seedlings to establish [25,29]. Numerous studies evaluating the
effectiveness of herbicide protection via activated carbon slurry banding (e.g., applying a strip
of activated carbon in crop rows) or seed technologies (e.g., activated carbon pelleting or coat-
ings applied directly to seeds) have been carried out over the last two decades [28,30–37].
However, reported success rates vary widely.

This reported variation in the performance of herbicide protection seed treatments may be
linked to the carbon application method (e.g., pellet, coating, banding), species-specific seed
traits, and their interactions [28,33–37]. Additionally, a high degree of variability in weather
and abiotic and biotic site conditions (e.g., elevation, topography, soil type, dominant plant
species, presence and depth of litter) is characteristic of the current state of the sagebrush
shrub-steppe biome. This variability may influence seedling establishment and impact the per-
formance of different herbicide protection treatments (or the herbicide itself) between sites
and experiments [38–40]. In order to advance and refine the use of herbicide protection in res-
toration, we need to better demonstrate its efficacy across multiple species and variable site
conditions. Coordinated, distributed, multi-year experiments provide an opportunity to assess
efficacy across variable site conditions and allow for adaptive refinement of restoration meth-
ods and seed treatments based on interim outcomes [41].

Here, we describe results of a coordinated series of trials conducted over three years, which
collectively evaluate the efficacy of different herbicide protection treatments (pelleting, coating,

banding) and formulations (pellet size, coating formulation, slurry or powder application) for native bunchgrasses and sagebrush across a geographically distributed array of field sites which were heavily invaded by exotic annual grasses. We also investigated whether seed delivery methods affected seeding success and whether the reduction of invasive annual grass litter prior to herbicide application is needed to increase herbicide contact with the seedbed and improve efficacy. Our goal was to more fully understand whether and how herbicide protection seed treatments can improve first-year native plant restoration outcomes in the sagebrush shrub-steppe. Specifically, we asked:

1. Does the fall application of pre-emergent herbicide:

    a. consistently reduce annual invasive species presence the following spring?

    b. consistently affect first-year seeding success of bare seeds of native species sown immediately prior to herbicide application?

2. How does first year seeding success (emergence and initial establishment) of herbicide protection seed treatments compare to one another and to untreated bare seed, when deployed into annual-invaded sites with and without herbicide?

    a. Do particular herbicide protection seed treatments perform better than others?

    b. Does the combination of herbicide protection treatment and reduced competition via herbicide application confer a net benefit over untreated seed?

3. How is seeding success modified by seeding method (broadcast vs shallow soil furrow)?

4. Does reducing litter depth before herbicide application modify the impacts of pre-emergent herbicide on annual invasive presence?

## Methods

### Experimental design overview

Three years of field experimentation took place at sites invaded with annual grasses. Some aspects of the experimental design (detailed below) varied adaptively from year to year, and others remained consistent (Tables 1 & 2). Each year, at three (2018) or five sites (2019, 2020), we tested two to four herbicide protection seed technology prototypes or seed treatments (henceforth, HP treatments) against untreated bare seed with and without herbicide application for multiple native, perennial species. For each year of experiments, all HP treatments and bare seed controls were seeded at the same time within each site, and all herbicide applications occurred immediately after seeding. We also tested different seed delivery and litter management methods, which we suspected could interact with seed treatment effects.

### Field sites and seeded species

All field sites formerly supported *Artemisia tridentata tridentata* var. *wyomingensis* (Wyoming big sagebrush) and perennial bunchgrass communities and were distributed across multiple states of the western United States (S1 Fig, Table 1). At the time of seeding, all sites were depauperate in native perennial species and highly invaded with one or more exotic annual grass species (ranging from 25–75% foliar cover). We used a climate partitioning approach described by Doherty et al. [42] to identify sites that maximally represented the climatic differences within the regions of the sagebrush biome that were characterized by Maestas et al. [43] and Chambers et al. [44] as having medium or low ecological resistance and resilience. Some sites became less suitable to testing after the first and second years, usually due to increases in

**Table 1. Field site information.**

| SITE | YEAR PLANTED | MGMT. | LOCATION | ELEV (m) | EXOTIC ANNUALS | OTHER SP |
|------|--------------|-------|----------|----------|----------------|----------|
| OR | '18, '19, '20 | private | 44.161° N, 120.32° W | 1119 | *Taeniatherum caput-medusae, Bromus tectorum, Sisymbrium altissimum* | *Poa bulbosa* |
| NV | '18, '19, '20 | state | 41.206° N, 116.87° W | 1717 | *B. tectorum, Erodium cicutarium, S. altissimum* | *Amsinkia sp.* |
| UT | '19, '20 | state | 41.649° N, 112.07° W | 1362 | *B. tectorum, T. caput-medusae, Lactuca serriola* | *P. bulbosa* |
| WY2 | '19, '20 | federal | 42.727° N, 108.36° W | 1748 | *B. tectorum, S. altissimum, L. serriola* | *P. secunda, Pascopyrum smithii* |
| WY1 | '18 | federal | 42.652° N, 108.67° W | 1875 | *B. japonicus, B. tectorum* | *Elymus trachycaulis, Achnatherum nelsonii* |
| ID1 | '19 | private | 43.189° N, 115.55° W | 1158 | *T. caput-medusae, B. tectorum* | *P. bulbosa, Chondrilla juncea* |
| ID2 | '20 | federal | 44.080° N, 116.57° W | 832 | *T. caput-medusae, B. japonicus, L. serriola* | *Epilobium brachycarpum, Amsinkia sp.* |

exotic perennial vegetation, and were replaced with more suitable sites in later years. Formal permitting to conduct work at three sites (ID2, WY1, WY2) was acquired via Categorical Exclusion permits from the Bureau of Land Management, with the remaining sites not requiring formal permitting. General site weather patterns were assessed by gathering site-specific, estimated, monthly data from PRISM Climate Data [45].

Four native species were used, with different species seeded in different years, but largely the same species used across sites within each year (Table 3). Species included the native shrub *A. tridentata tridentata* var. *wyomingensis* (hereafter, ARTR), and three native perennial bunchgrasses: *Elymus elymoides* (bottlebrush squirreltail; hereafter, ELEL), *Poa secunda* (Sandberg bluegrass; hereafter, POSE), and *Pseudoroegneria spicata* (bluebunch wheatgrass; hereafter, PSSP). In 2018 and 2019, different seed sources of each species were seeded at each site (the most local provenance available), whereas in 2020 a consistent seed source of each species was used across all sites (S1 Appendix, Table 1).

## Production and preparation of seed and seed treatments

Three classes of herbicide protection treatments were tested in our study: 1) multi-seed herbicide protection pellet (HP pellet), tested in all three years for all species; 2) single-seed herbicide protection coatings (HP coatings), tested only in 2020 and only for grasses; and 3) activated carbon banding in which a dry powder (2019) or wet slurry (2020) was applied over already-sown seed (HP banding), tested only for ARTR (Tables 3 & 4). Small and large sizes of HP pellets (relative to the size of the grass seed, which varied by species [37]) were tested on

**Table 2. Experimental factors.**

| YEAR | FACTOR | NAME | VALUES | DEFINITION |
|------|--------|------|--------|------------|
| All | Field site | SITE | OR, NV, WY, ID, UT | Experimental field site, see Table 1 |
| All | Seeded species | SPEC | ARTR, ELEL, POSE, PSSP | Native species seeded, see Table 3 and S1 Appendix |
| All | Seed technology treatment | TECH | bare seed, HP pellet (large, small), HP coating (commercial, vortex), HP band | Herbicide protection seed treatment applied, see Table 3 |
| '18, '19 | Seed delivery method | DELIV | broadcast, furrow | Seed broadcast uniformly to entire subplot, or added only into shallow furrows in soil surface |
| All | Herbicide application | HERB | herbicide applied, no herbicide | Herbicide applied soon after seeding, or nothing applied |
| '19 | Litter management | LTR | reduced, intact | Litter reduced via hand-raking, or litter left intact |

**Table 3. Experimental design.**

| Year seeded | Sites | Species | Herbicide | HP seed treatment | Delivery method | Litter mgmt. | Exp. Design | Exp. unit | Seed rate | Replicates |
|---|---|---|---|---|---|---|---|---|---|---|
| **2018** | OR[a] | POSE | •Yes[b] | •Bare seed | •Broadcast | •Reduced | Fully randomized, complete factorial | 1 x 1 m plot | $\frac{(400 \ PLS^g)}{m^2}$ | 5 |
| | NV[a] | ELEL | •No | •HP pellet—L | •Furrow | | | | | |
| | WY1[ac] | ARTR | | •HP pellet—S | | | | | | |
| **2019** | OR[a] | POSE | •Yes | •Bare seed | •Broadcast | •Reduced | Fully randomized, partial factorial | 1 x 1 m plot | $\frac{(400 \ PLS)}{m^2}$ | 5 |
| | NV | ELEL | | •HP pellet—L | •Furrow | •Intact[f] | | | | |
| | WY2[a] | ARTR | •No | •HP pellet—S | | | | | | |
| | UT[a] | | | •AC band—dry[d] | | | | | | |
| | ID1 | | | | | | | | | |
| **2020** | OR[a] | PSSP | •Yes | •Bare seed | •Furrow | •Intact | Randomized split-plot, partial factorial | 1 x 0.5 m subplot | $\frac{(572 \ PLS)}{m^2}$ | 7 |
| | NV | ELEL | | •HP pellet—S | | | | | | |
| | WY2[a] | ARTR | •No | •HP coat—Kamterter | | | | | | |
| | UT | | | •HP coat—vortex | | | | | | |
| | ID2 | | | •HP band—slurry[e] | | | | | | |

Experimental factors and factor levels tested in each year-by-site trial are shown left of vertical line, and experimental details are right of vertical line. Within each year, unless noted, all levels of each multi-level factor were combined factorially with all other multi-level factors, and single-level factors were applied uniformly to all other factor levels. New sites were eventually used in WY and ID (marked with 2) when original sites (marked with 1) became less suitable over time.

[a]Herbicide mix also included 438 ml/ha formula Accord (219 ml/ha AI glyphosate) at these sites and years.

[b]Lower rate of 584 ml/ha formula Plateau (137 ml/ha AI imazapic) used in 2018; remaining years used higher rate (730 ml/ha formula, 172 ml/ha AI).

[c]WY site used ARTR in 2018 instead of POSE and ELEL that were used at NV and OR sites.

[d]Used in place of large HP pellet for ARTR.

[e]Used instead of HP coatings for ARTR, due to unavailability of coatings for this species.

[f]Additional unseeded plots were added to test intact litter vs reduced litter in 2019; all seeded plots had litter reduction treatment.

[g]PLS = pure live seed; an estimate of the number of viable seeds within a sample that accounts for the nonviable fraction of seed lot.

each species in 2018 and 2019, with only the small size tested in 2020. All HP pellet production followed Madsen et al. [46] and Baughman et al. [37] and involved extruding an activated carbon dough through a die to form pellets (Table 4). Two prototypes of HP coatings were tested for each of two grass species in 2020: a commercially made, high-integrity coating produced in a traditional rotating seed coater by Kamterter Products, LLC (Waverly, Nebraska, USA), and a lower-integrity coating (vortex) that was produced in-house as described by Holfus et al. [47]. Finally, two methods of HP banding were used on ARTR only: a powder band (comprised of the same ingredients and ratios as HP pellets) in 2019, and a 14:1 slurry of water and pure activated carbon in 2020. All tested seed treatments used the same active ingredient, Darco Grosafe activated carbon (Cabot Corporation, Boston MA, USA). Additional seed treatment production details can be found in S1 Appendix.

Within each year, target viable seeding rates were identical regardless of species or seed treatment or site and were calculated as pure live seed (PLS, or the number of viable seeds estimated in the sample). Rates were 400 PLS/m$^2$ (2018, 2019), or 572 PLS/m$^2$ (2020). Each sample to be seeded was measured to the target PLS by weight using estimates of PLS per bulk gram for each species and seed treatment (S1 Appendix).

## Site preparation and seeding

In 2018 and 2019, the experimental setup at each site consisted of a completely randomized, full factorial design of 1 x 1 m plots with five replicates (Table 3). In 2020, the experimental

**Table 4. Seed treatment specifications.**

| SEED TECHNOLOGY | SPECIES | YEAR | SPECIFICATIONS | INGREDIENTS |
|---|---|---|---|---|
| HP pellet—large | ELEL | '18, '19 | 8 x 16 x 16 mm pod; 2 g; ~8 viable seeds per unit | 42–44% Pelbon bentonite clay, 33–34% Darco GroSafe activated carbon, 13–14% Deschutes Recycling Biofine compost*, 6% California Vermiculture worm castings compost*, 0.2–3% seed |
| | ARTR | '18 | 8 x 8 x 16 mm pellet; 1.1 g; ~8 viable seeds per unit | |
| | POSE | '18 | | |
| HP pellet—small | ELEL | '18, '19, '20 | 8 x 8 x 16 mm pellet; 1.1 g; ~8 viable seeds per unit | |
| | PSSP | '20 | | |
| | ARTR | '18, '19, '20 | 4.5 x 4.5 x 9 mm pellet; 0.3 g; ~8 viable seeds per unit | |
| | POSE | '18, '19 | 6.3 x 6.3 x 12 mm pellet; 0.6 g; ~8 viable seeds per unit | |
| HP band—dry | ARTR | '19 | 164 g dry powder added per 4 m of seeded furrow | Same as HPP, but 0% seed |
| HP band - slurry | ARTR | '20 | 320.5 g slurry added per 3 m of seeded furrow | 6.4% Darco GroSafe activated carbon, 93.6% water |
| HP coat—Kamterter | ELEL | '20 | 25 mg coating; 1 seed per unit | 48% Darco GroSafe activated carbon, 52% proprietary binders |
| | PSSP | '20 | | |
| HP coat—vortex | ELEL | '20 | 18-20mg coating; 1 seed per unit | 99–99.16% Darco Grosafe activated carbon, 0.84–1% selvol |
| | PSSP | '20 | | |

*Sieved with 1.9 mm (#16) mesh.

setup at each site consisted of seven split-plots (split = herbicide application), with each split containing a single replicate of each unique treatment in 1 x 0.5 m subplots. Litter was reduced via hand raking (hereafter, litter reduction treatment) on all plots in 2018 and all but the litter-intact plots in 2019, but did not continue in 2020, based on our results from 2019. Subplots were uniformly seeded by hand, either in the furrow only (with no burial), or over the entire subplot area for the broadcast treatment (half of all plots in 2018 and 2019). For subplots receiving the furrow seed delivery treatment (half of all plots in 2018 and 2019, all subplots in 2020), 2–2.5 cm deep and 2.5–3 cm wide furrows were made using hand tools just prior to seeding. Immediately after seeding, subplots assigned to the carbon strip treatment received the dry or wet carbon applied in a 2 cm wide strip down the furrows via a plastic condiment. Seeding occurred mostly in October, but ranged from September to November due to logistical constraints (S1 Appendix, Table 2).

Within 1–2 days of seeding, an herbicide solution was applied via backpack sprayer to all plots or split-plots receiving the herbicide treatment, with a 0.5–1.0 m overspray buffer (S1 Appendix). Nothing was applied to plots not receiving the herbicide treatment. Herbicide was applied at the target rate of 584 ml/ha (8 oz/ac) formula of the pre-emergent Plateau (137 ml/ha AI of imazapic salt) in the 2018 seeding year, and 730 ml/ha (10 oz/ac) formula of Plateau (172 ml/ha AI) in 2019 and 2020. For some sites and years, 438 ml/ha (6 oz/ac) formula of the nonselective broadleaf Accord (219 ml/ha AI glyphosate) was added to target occasional green-up of some weed species (Table 1). In ID in 2019, mixing errors occurred which resulted in a 4-fold higher than intended rate of pre-emergent (741 ml/ha AI). We deployed germination bags containing bare seed samples to estimate field germination rates of seeded species at each site for all three years. Additional site preparation, seeding, and germination bag details can be found in S1 Appendix.

## Data collection

We quantified the density of seeded species and the density and cover of non-seeded, onsite vegetation two times during the first growing season (April–June) of each year (S1 Appendix, Table 2). We focused on testing the effect of seed treatments on seedling performance during the first growing season because this period represents the most consequential bottleneck in native bunchgrass seedling establishment [48]. The two sampling events each year were chosen to correspond with the period of peak seedling density during the early growing season (hereafter, early seedling count) and the later-season period in which living seedling density were lower but more representative of first-season establishment, before summer dormancy (hereafter, late seedling count). Early monitoring in 2020 was prevented for the experiment seeded in 2019 due to the COVID-19 pandemic.

We measured the density of all seeded species in all plots or subplots, regardless of whether they were seeded with those species. This was done to detect any notable background density of species seeded in our trials, and few to none were found (S1 Appendix). The density of seeded species was measured in the entire seeded subplot area for every sampling event, and seedlings found during the first monitoring were marked with toothpicks (S1 Appendix). Seeded species height and leaf count were measured in the late sampling only, taken from the three seedlings in each subplot closest to the bottom left corner of the sampling frame.

Onsite invasive annual grass (IAG) and invasive annual forb (IAF) density was measured using a site-adjusted, downscaled sampling area method to balance count accuracy with large differences in density among groups and sites (S1 Appendix). Percent foliar cover of IAG and IAF was visually estimated to the nearest percent across the entire seeded area for every group. Cover was also recorded for two non-plant groundcover categories: litter and bare ground.

## Analysis

Response variables of seeded species for each sample unit were early and late season seedling density, as a percentage of viable seed sown (Table 2; hereafter, seedling count), and average late season height and leaf count. Response variables of other vegetation were plant density and foliar cover in the subplot for IAG and IAF. Experimental factors consistent to all years included field site (SITE), species (SPEC), herbicide application (HERB), and herbicide protection seed treatment (HPTRT). Factors used in only one or two years included seed delivery method (DELIV) and litter management (LTR).

Separate models were used for each year of data, as described below, due to the year-to-year differences in experimental design and tested factors resulting from the adaptive nature of our research questions and experiments. Additionally, within each year, multiple models were often needed to separate the partially factorial full experimental design into fully factorial components. For example, specific seed treatment prototypes were often different for ARTR and grasses. Finally, to clearly assess effects of herbicide on the bare seed treatment specifically (question 1), each year's model (described below) was run again on a reduced dataset that contained only the bare seed treatment. Each model run was a fully factorial ANOVA with two or more factors performed in JMP (SAS Institute, Carey, NC), with $p < 0.05$ deemed significant. Significant model effects and interactions were investigated with post-hoc Tukey HSD tests.

For the 2018 seeding trials, the SITE factor was highly influential and involved in a significant five-way interaction when included in the full five-factor model, so an identical four-factor model was run independently for each level of the SITE factor to improve interpretability, and included SPEC, HPTRT, HERB, DELIV, and all factorial interactions. Seedling height and leaf count responses had too many missing data (from subplots with no seedlings) to examine 3 or 4-way interactions in OR or NV, or even two-way interactions in WY.

For the 2019 seeding trials, the main model included SITE, SPEC, DELIV, HPTRT, HERB, and all interactions. The effect of DELIV on seeded species responses was consistent (described in the results; poorer success in the broadcast than the shallow furrow treatment, at times too low for analysis), though the effects of other factors interacted differently for broadcast than shallow furrow delivery. To limit our inference to the delivery treatment with the most success, and reduce the zero-inflation of our data, DELIV was removed from the final model, along with all data from broadcast-seeded plots, for analysis of seedling count and the density and cover of IAG and IAF vegetation. Seeded species height and leaf count had too many missing data for some sites and species to run in this model and were analyzed in separate two factor models (HPTRT, HERB, interaction) for each combination of SITE and SPEC with enough data for analysis (ELEL in OR, UT, and ID; ARTR and POSE in OR). Additionally, in 2019, the effect of litter reduction on the cover and density of invasive annual species and other on-site vegetation as well as ground cover was examined using a three-factor model with SITE, HERB, LTR, and interactions. This model contained only unseeded data, and interpretation focused solely on the main or interactive effects of the LTR factor.

For the 2020 seeding trials, as in 2018, SITE was highly and complexly influential on seeded grass and other vegetation responses, so a three-factor model was run separately for each site, which included SPEC, HPTRT, HERB, and all interactions. This model was used to analyze early and late seedling count, seedling height, and seedling leaf number for grass species only. Seeding responses for ARTR were analyzed with a separate three-factor model including SITE, HPTRT, HERB, and interactions, though data from WY, UT, and NV were removed because no ARTR emerged in those sites.

## Results

### Summary of weather and germination trends during the experiments

During the two seasons most influential to germination and seedling emergence in each planting year, Dec–Feb (winter) and Mar–May (early spring), several general trends in weather were observed over the three planting years. Both winter and spring of the 2018 seeding year were generally cooler (0.25–1.4 C) and wetter (up to 50%) than normal at all sites (S2 Fig). Conversely, drier than normal winter and spring conditions were dominant after the 2019 and 2020 plantings across all sites (10–75% drier), and warmer than average winters (0.1–1.25 C) was the most common trend for temperature. Cumulative germination of bare seeds recovered from shallow germination bags (as a percent of estimated viable seeds sown) varied by species, site, and year, and ranged from 25% to 100% (S2 Appendix, S3 Fig).

### Effects of herbicide application on invasive annual species

In general, fall application of pre-emergent herbicide resulted in a notable reduction in the spring density of IAG in all but one site in one year (2020 UT; Fig 1), as well as a large reduction in spring IAF density in all but three sites in one year (2019 UT, OR, and WY; though WY had almost no IAF). In 2018, herbicide application reduced IAG and IAF density at all sites by 47–89% and 67–97% (HERB main effects; $F_{1,108} = 14$–$36$, $p < 0.005$; $F_{1,108} = 5$–$59$, $p < 0.017$). In 2019, herbicide application reduced IAG density by 58–99% (average 82%) at all sites (SITE*HERB interaction; $F_{4,928} = 4252$, $p < 0.001$), and the effect of herbicide on IAF varied by site, reducing density by 93–99% in NV and ID, while being associated with 24% higher IAG density in UT (no effect in other sites; SITE*HERB interaction; $F_{4,928} = 762$, $p < 0.001$). In 2020, herbicide application reduced IAG density by 90% (WY and NV) and 99–100% (OR and ID), with no effect in UT (SITE*HERB interaction; $F_{4,774} = 73.1$, $p < 0.001$), and reduced IAF density by 78–99% at all sites in which IAF were present (all but WY, which had no IAF;

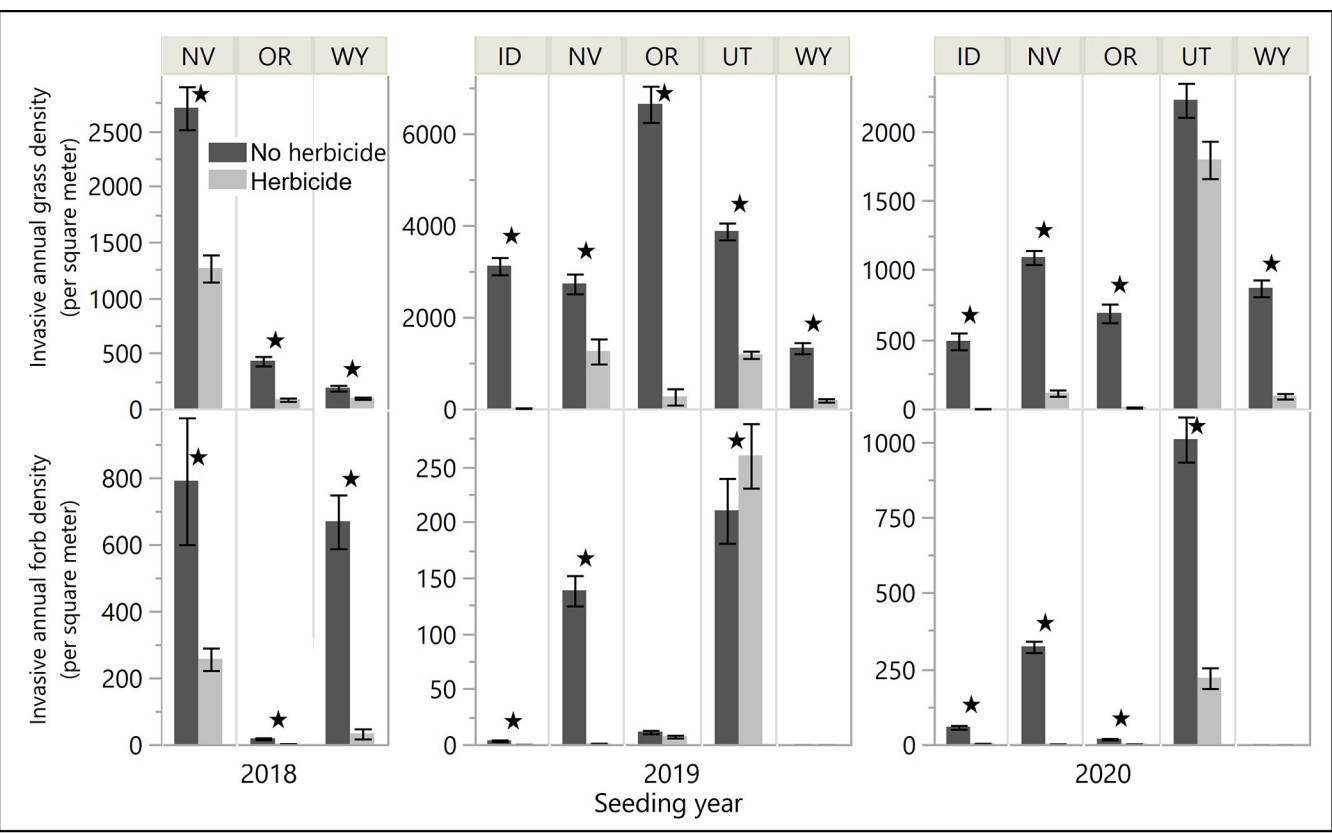

**Fig 1. Herbicide effects on invasive annual vegetation.** Density, per square meter, of living invasive annual grasses (IAG, top) and forbs (IAF, bottom) by seeding year (across the bottom) and experimental site (across the top) late in the first growing season (May-Jun) after application of pre-emergent herbicide the previous fall (Sep-Nov), in areas that received the herbicide (light gray) compared to those that did not (dark gray). Rate of herbicide application was 584 ml/ha (8 oz/ac) formula of Plateau (137 ml/ha AI imazapic) in the 2018 seeding year, and 730 ml/ha (10 oz/ac) formula (172 ml/ha AI) in the remaining years. Black stars indicate significant difference among the herbicide treatments. Error bars are standard errors. Note differences in scale for each year.

SITE*HERB interaction; $F_{4,773} = 95.1$, P < 0.001). The effect of herbicide application on foliar cover of IAG and IAF generally tracked those of plant density, with only a few exceptions (S2 Appendix for additional results).

## Effects of herbicide application on seeding outcomes of bare seed

Across the three seeding years, there was only sparse and mixed evidence that bare seed sown into exotic, annual-invaded sites differed in seedling count or size in the presence vs the absence of pre-emergent herbicide (specific results by year are below). This was counter to our hypotheses that bare seed success would consistently be reduced by the presence of herbicide. Although neither seeding scenario represented a common restoration practice (seeding bare seed into fully-invaded areas without herbicide pre-treatment; or applying herbicide immediately after seeding with bare seed), this comparison was helpful for understanding whether the larger barrier to seeding success was toxicity from herbicide (indicated by reduced success in the presence of herbicide) or competition with invasive species (indicated by reduced success in the presence of competition in the absence of herbicide).

In 2018 and 2019, for bare seed, early and late seedling count (Fig 2) and size (S4 Fig) did not differ in the presence vs the absence of herbicide at any site or for any species. No significant main or interactive effects of HERB were present for any site in 2019 or for NV or WY in

2018, and none of the several significant interactions involving HERB in OR in 2019 demonstrated differences between herbicide treatments (late seedling count: SPEC*HERB*DELIV interaction, $F_{1,32}$ = 5.38, p = 0.027; seedling height: SPEC*HERB*DELIV interaction; $F_{1,17}$ = 4.87, p = 0.041; S2 Appendix). In 2020, there were two instances of lower late seedling count in the presence vs the absence of herbicide (90% reduction in ID for ELEL only [$F_{2,30}$ = 3.58, p = 0.04], and 63% reduction in OR regardless of species [$F_{1,32}$ = 12.2, p = 0.001]), as well as two instances of larger seedling size in the presence vs the absence of herbicide (50% higher leaf count in NV regardless of species [$F_{1,17.4}$ = 16.2, p = 0.001], and 78% taller height in UT for PSSP only [$F_{1,9.3}$ = 13.4, p = 0.004]) (S4 Fig). In 2020, an interaction involving HERB for late season seedling count in WY demonstrated no effect of HERB for any of the three species involved (SPEC*HERB interaction; $F_{2,28.3}$ = 3.91, p = 0.032; S2 Appendix).

## Comparison of seeding outcomes among herbicide protection treatments, with and without herbicide

Detailed results describing all differences in seeding outcomes among the various HP treatments tested in each year, with and without herbicide, are provided in S2 Appendix, and summarized below by year. Differences among HP treatments and bare seed are given in the next subsection.

In 2018, the large and small pellet generally performed similarly, with a few instances of the large pellet demonstrating improved outcomes. The large HP pellet showed 1.5–2.5-fold higher early seedling count than small HP pellet in NV for POSE in the presence of herbicide (SPEC*TRT*HERB interaction; $F_{2,96}$ = 3.1, p = 0.05), and in WY for all ARTR (with and without herbicide; SPEC*TRT interaction; $F_{2,96}$ = 12.1, p < 0.001), though no differences remained by the late season count at either site (Fig 3). The large pellet was associated with 38–50% higher seedling height and leafiness for all species in OR in the presence of herbicide (S5 Fig; TRT*HERB interactions; $F_{2,73}$ = 3.5–5.7, p < 0.033).

In 2019, the large pellet for both grass species, regardless of site, showed 2.1-fold higher mean late season seedling count than small pellet in the absence of herbicide, with no such differences in the presence of herbicide (Fig 3; HPTRT*HERB interaction; $F_{2,240}$ = 4.22, p = 0.0159). For ARTR in 2019, the small HP pellet and HP band treatments showed no differences in seedling count, though the pellet was associated with 2.1-fold taller seedlings in the presence of herbicide at the OR site (S5 Fig; HPTRT*HERB interaction, $F_{2,12}$ = 3.90, p = 0.0493).

In 2020, there were no differences in early or late seedling count between any HP treatments for ARTR at any site, for either grass species in NV and WY, or for PSSP in ID (Fig 4). All differences in seedling count among treatments in 2020 were regardless of herbicide treatment. The small HP pellet had 1.4-2-fold higher seedling count than one or both coatings for both grass species in OR and UT and ELEL in ID, but this effect only persisted to late season count for ELEL in ID (S2 Appendix). The small pellet was associated with a 2.6-fold higher leaf number over Kamterter coatings for PSSP in ID and 40–50% taller seedlings than both HP coatings for both grass species in NV, with both effects in the presence (but not absence) of herbicide (S5 Fig). Differences among the two coatings were rare, with vortex producing higher seedling count than Kamterter coating only for ELEL in ID and only early in the season (Fig 4), and higher leaf numbers than HP pellet and Kamterter coatings for both species in NV (S5 Fig).

## Performance of herbicide protection treatments compared to bare seed with and without herbicide

This section focuses only on instances in which any of the HP seed treatments were associated with different outcomes than the bare seed treatment. Details about the effects of herbicide

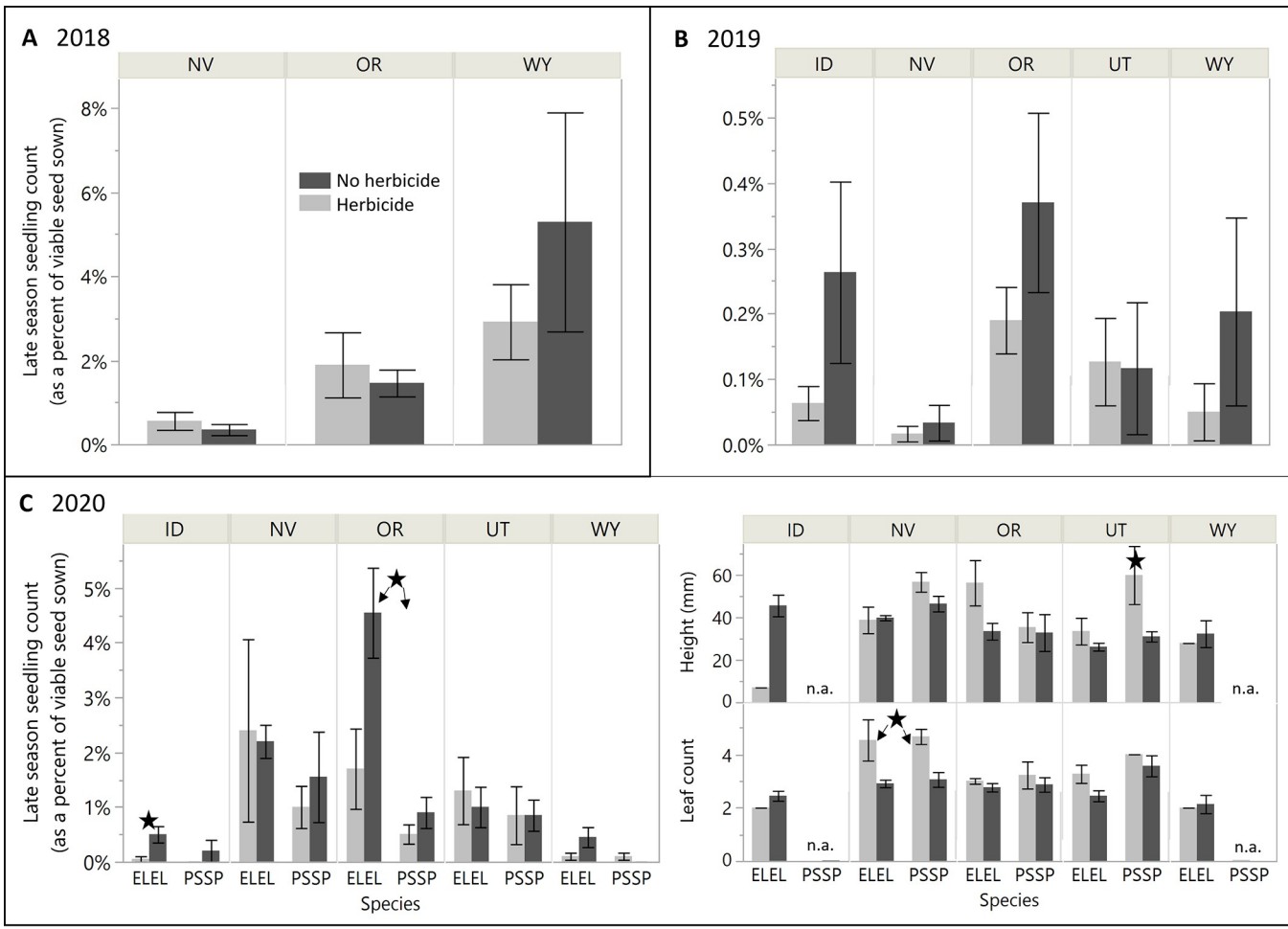

**Fig 2. Herbicide effects on bare seed treatment.** Observed differences in late season seedling count and seedling size from bare, unprotected seed sown in the presence (light gray) and absence (dark gray) of pre-emergent herbicide applied immediately after sowing, by site and across species in seeding year 2018 (panel A) and 2019 (panel B), and by site and species in seeding year 2020 (panel C). Only data from the shallow furrow delivery method are shown. Black stars indicate significant difference among the herbicide treatments (P < 0.05), with arrows added in panel B to indicate herbicide effects that were regardless of seeded species identity. Error bars are standard errors.

presence on seed treatments as well as differences in seeding outcomes among the HP treatments themselves are given in the two prior subsections.

In 2018, differences in seedling count and seedling size between herbicide protection and bare seed treatments were generally mixed, and varied by species, herbicide application, site, and season. There were no differences in early or late seedling count for ELEL (Fig 4A). For POSE, both the small and large pellet showed 2.3–2.7-fold higher early count than bare seed at both sites; SPEC*HPTRT interaction; $F_{2,96} = 11.7$, $p < 0.001$). By late season, neither grass species showed differences in seedling count related to seed treatment. The only differences in late season seedling size were that large HP pellet had 60% taller seedlings than bare seed (regardless of site, species, herbicide; $F_{2,73} = 5.7$, $p = 0.005$), and bare seed of both species had nearly twice as many leaves as small HP pellet in OR, in the presence of herbicide (S5 Fig; HERB*HPTRT interaction; $F_{2,73} = 3.54$, $p = 0.034$). Seedling count for ARTR was 48–78% lower for both sizes of HP pellet than bare seed in both the early and late season (Fig 4A).

In 2019, differences between seed treatments and bare seed were more uniform than in 2018 but varied by herbicide presence. In the presence of herbicide, both pellet sizes were

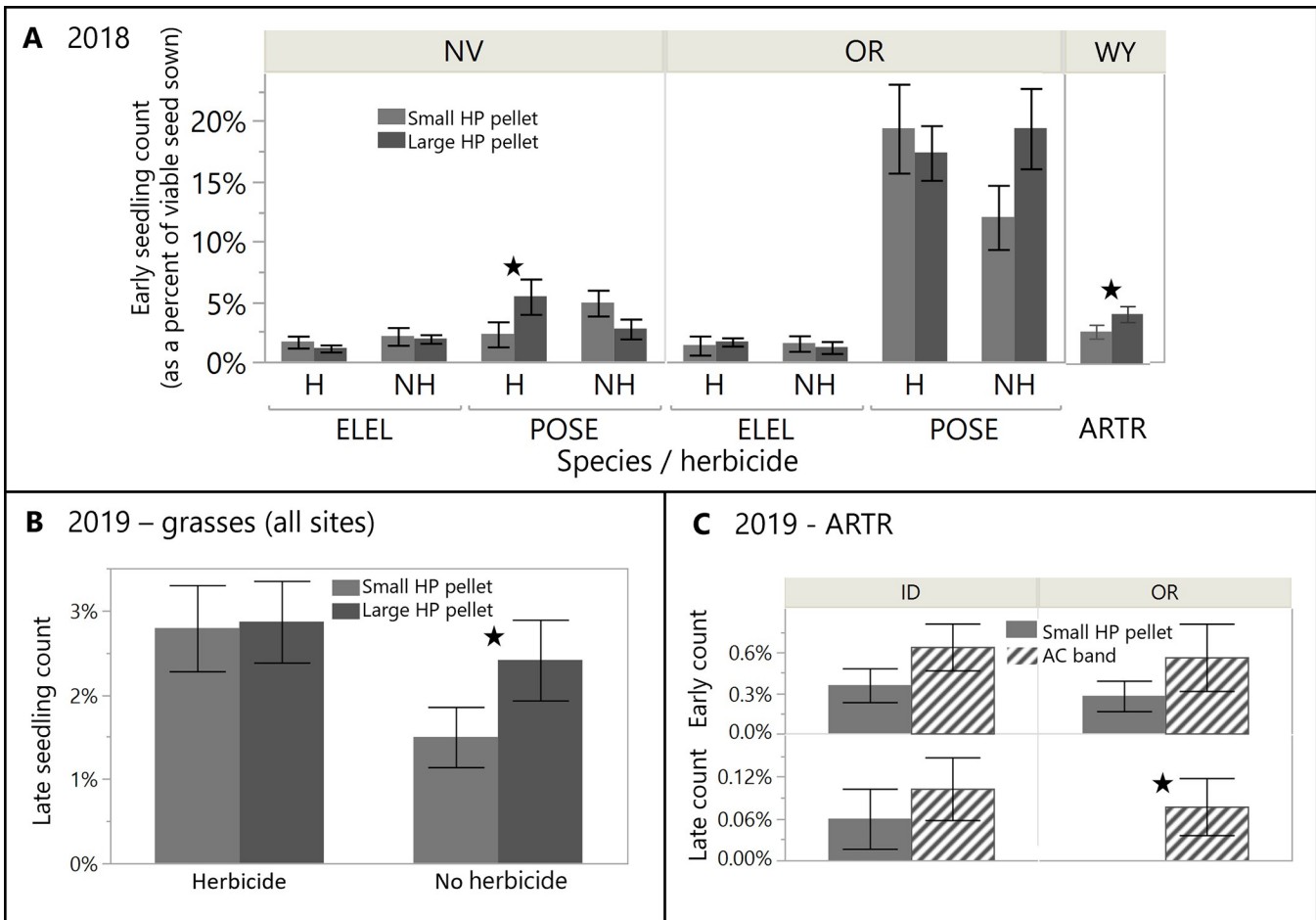

**Fig 3. Comparison of HP seed treatments.** Observed differences among HP seed treatments (pellets, coatings, HP band) in both early and late season seedling count (as a percent of viable seed sown), in 2018 (panel A) by seeding site, species and herbicide treatment (H = herbicide, NH = no herbicide), in 2019 for grasses (panel B) by herbicide treatment, and in 2019 for sagebrush (ARTR, panel C) by site. Refer to Fig 4 for significant differences among HP seed treatments in 2020. All data shown used the shallow furrow delivery method, with all broadcast delivery data removed from the two years it was used (2018, 2019). Black stars indicate significant difference among the HP seed treatments (P < 0.05). Error bars are standard errors.

associated with 2.7-3-fold higher late season grass seedling count than bare seed, regardless of site or species, whereas neither pellet size differed from bare seed seedling count in the absence of herbicide (Fig 4; HPTRT*HERB interaction; $F_{2,240}$ = 4.22, $p$ = 0.0159). Large and small pellet were also associated with 1.9-fold taller seedlings for ELEL than bare seed, in the presence, but not in the absence of herbicide (S5 Fig; HPTRT*HERB interaction, $F_{2,16}$ = 5.49, $p$ = 0.0153). There were no effects of seed treatment on seedling count for ARTR, though carbon strip was associated with shorter seedlings (12.2 mm) than bare seed (5.8 mm) and small HP pellet (27.5 mm) in the presence of herbicide, with no such difference in the absence of herbicide (S5 Fig; HPTRT*HERB interaction, $F_{2,12}$ = 3.91, $p$ = 0.0493).

In 2020, differences in seedling count among seed treatments for grasses varied notably by site. Full details are available in S2 Appendix and are summarized here. No differences were present in WY. The most common difference across other sites was that small HP pellet was associated with 1.5-4-fold higher seedling count than bare seed in the early season (Fig 4; ELEL in NV, both species in OR and ID), with few differences remaining by the late season (only ELEL in ID), and only one instance of these differences occurring only in the presence of

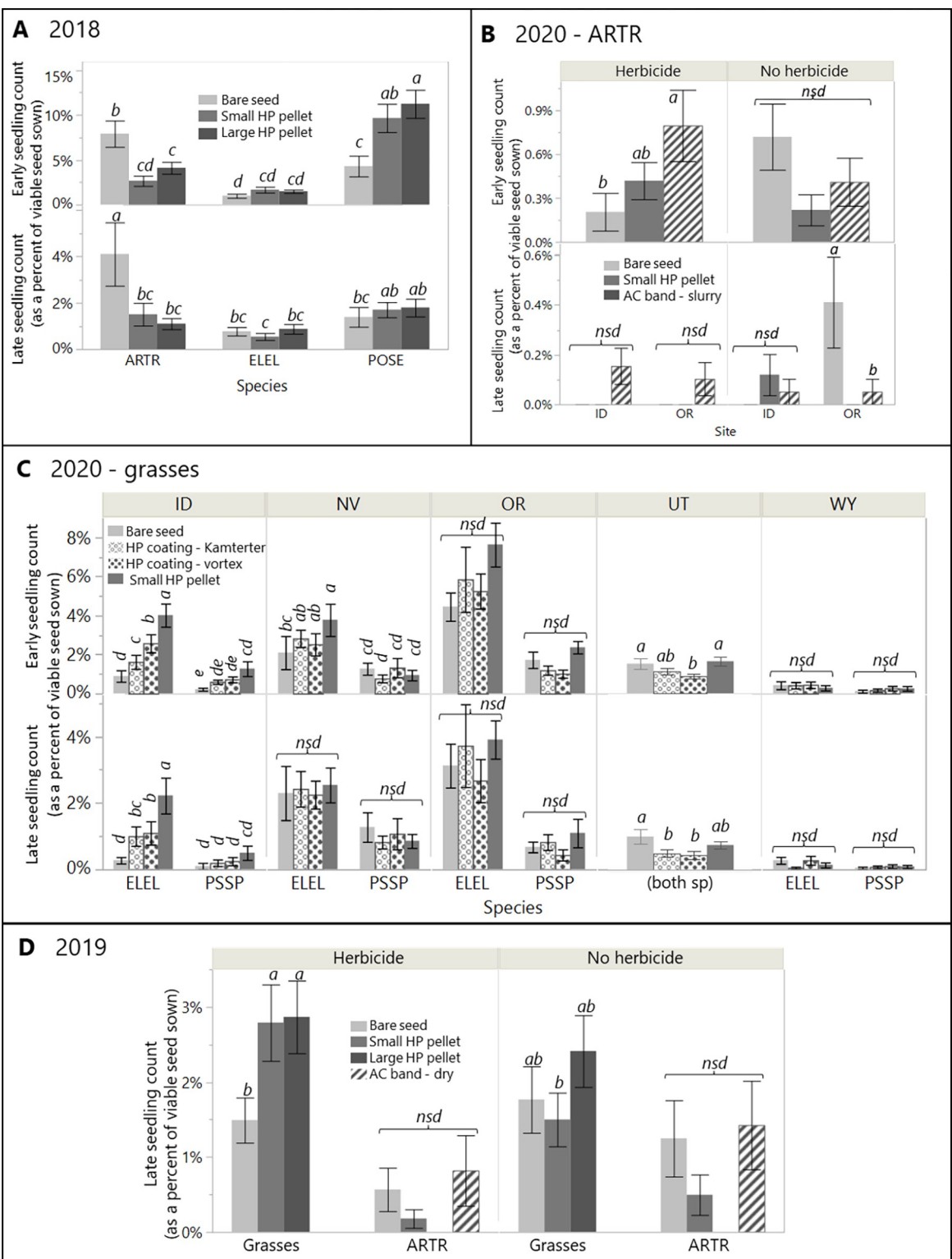

**Fig 4. Comparison of HP and bare seed treatments.** Observed differences among HP seed treatments (medium gray, dark gray, and patterned bars) and bare, untreated seed (light gray) in both early and late season seedling count (as a percent of viable seed sown), in 2018 (panel A) by species, in 2019 (panel D) by species type and herbicide treatment, in 2020 for grasses (panel C) by site and species and in 2020 for sagebrush (ARTR, panel B) by site and herbicide treatment. All data shown used the shallow furrow delivery method, with all broadcast delivery data removed from the two years it was used (2018, 2019). Bars that do not share the same letter are significantly

different as determined by post-hoc Tukey HSD test (P < 0.05), with separate tests conducted for each response for each year (and for each site for 2020 grasses). NSD = no significant differences between seed treatments for the comparison shown. Error bars are standard errors.

herbicide (ELEL in NV). Instances of either coating producing higher seedling count than bare seed were limited to ELEL in ID, where both coatings were associated with 2–3-fold higher early and late count than bare seed. In UT, bare seed of both grass species was associated with 1.5–2 -fold higher seedling count than one or both coatings, in the early (vortex) and late (both) season, as well as 17–32% taller seedlings than HP pellet and vortex coating, regardless of other factors (S5 Fig).

In 2020, ARTR plots of any treatment had very few seedlings (especially in NV, WY, and UT which were omitted), limiting analysis. The HP band treatment was associated in a 4-fold higher early seedling count in the presence (but not the absence) of herbicide, regardless of site, but by late season this pattern persisted only at the OR site (Fig 4). Additionally, in OR, bare seed had 5-fold higher seedling count than HP band and HP pellet in the absence of herbicide.

## Effects of seeding method on seeding outcomes

In 2018, broadcast seeding was never associated with higher early or late seedling count, seedling height, or leaf numbers than shallow furrow seeding in any comparison across sites or species (Fig 5). Shallow furrow seeding generally had greater seedling count and size than broadcast seeding, and the limited but complex exceptions are detailed in S2 Appendix.

In 2019, shallow furrow seeding was associated with higher late season seedling count than broadcast seeding across sites and species, regardless of seed treatment or herbicide application, and this effect was stronger for ELEL (12-fold higher) than POSE (11-fold) (Fig 5; DELIV*SPEC interaction; $F_{1,479} = 47.5$, $p < 0.001$). Seedling count of ARTR showed a similar trend at the only site with enough seedlings to analyze (OR), with 8.5-fold higher seedling count for bare seed and HP pellet in furrows than broadcast, regardless of other factors ($F_{1,32} = 7.45$, $p = 0.0102$).

## Effect of litter reduction on pre-emergent herbicide effectiveness against annual invasive weeds

Overall, the 2019 litter reduction treatment consistently reduced litter depth and cover but did not modify the effects of herbicide application on invasive annual plant density and cover. The litter reduction treatment consistently reduced litter depth from 18–45 mm to 7–11 mm, with the strongest reductions in OR and WY and the weakest in NV and UT (S2 Appendix; SITE*LTR interaction; $F_{4,79} = 7.1$, p < 0.001;). The litter reduction treatment had no effect on the density or cover of IAG the spring following herbicide application (LTR main effects; $F_{1,80} = 0.2–0.8$, $p > 0.35$), nor did it change how density or cover of IAG was affected by herbicide (LTR*HERB interactions; $F_{4,80} = 3.1–3.9$, $p > 0.051$), regardless of site. All but the UT and OR site showed no main or interactive effects of litter reduction on IAF cover or density (SITE*LTR and SITE*LTR*HERB interactions; $F_{4,19–80} = 3.6–4.5$, $p < 0.009$). In UT, there was a higher density of IAF when litter was reduced (190 plants m$^{-2}$) than when left intact (70 plants m$^{-2}$), regardless of herbicide application, with no such differences in cover. In OR, the overall low IAF cover was lower for reduced litter (0.2%) than intact litter (2.8%) in the presence of herbicide.

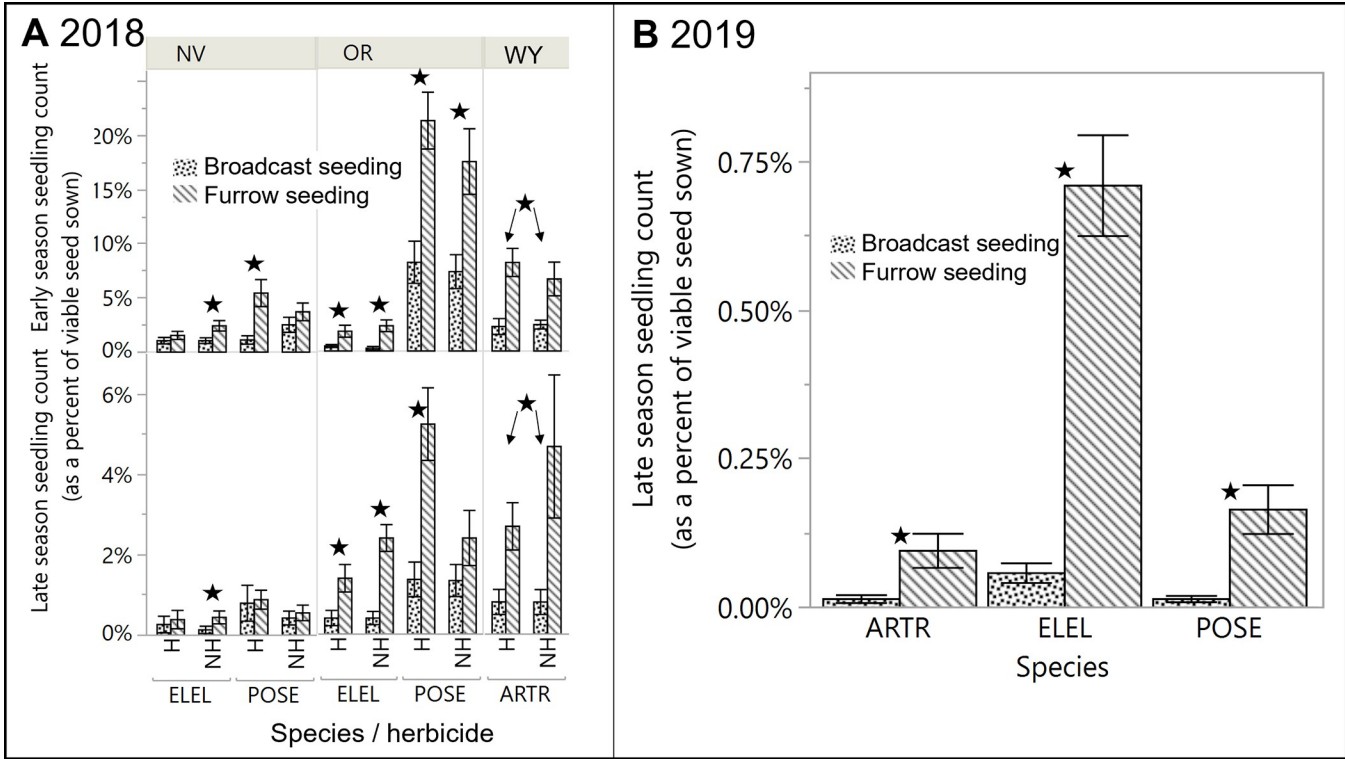

**Fig 5. Seeding method effects.** Observed differences between broadcast and shallow furrow seeding delivery in early and late season seedling count (as a percent of viable seed sown) in 2018 (panel A) by species and herbicide treatment (H = herbicide, NH = no herbicide) and 2019 (panel B), by species. Black stars indicate significant differences between seeding delivery treatments (P < 0.05), with arrows added in panel B to indicate a delivery effect regardless of herbicide treatment. Error bars are standard error.

## Discussion

The intended purpose of herbicide protection seed treatments is to improve seed-based restoration outcomes by minimizing competition with invasive annual species using herbicide while simultaneously protecting the seed of desired species from herbicide toxicity. In this series of trials, herbicide applications largely resulted in notable reductions of the targeted exotic, annual species, though near-complete reduction of invasive annual grasses ($\geq$ 99% reduction in density or cover) was only achieved in two of the 13 trials (Fig 1). However, we did not find a consistently negative effect of pre-emergent herbicide on unprotected bare seed of the restoration species (Fig 2), as was expected and reported elsewhere [49]. That is, for bare seed, late season seedling count was lower in the presence of herbicide than the absence of herbicide in only 3 of 12 trials for grasses and 2 of 13 of trials for ARTR, and, conversely, was higher in the presence of herbicide just as frequently (3 of 12 trials) for grasses. Sheley et al. [50] also found inconsistent effects of imazapic on seeded perennial establishment across many factors (species, sites, herbicide rates, and application dates). This lack of a consistent, negative effect of herbicide on unprotected seeds is likely related in part to the equally challenging establishment conditions in our no-herbicide controls (which had high levels of invasive vegetation; discussed below), and often hampered our ability to assess the protective ability of the tested technologies.

Overall seeding success was low, likely due to challenging, post-germination abiotic conditions; however, there were numerous differences in seeding outcomes among the herbicide protection treatments as well as between these treatments and bare seed. In the two years in

which multiple sizes of HP pellet were compared, the large pellet was associated with better performance (early seedling count and late season seedling size in 2018; late season seedling count in 2019) relative to the small pellet in both the presence and absence of herbicide. These findings run counter to the only other existing study comparing HP pellet sizes [37], which found overall improved performance of the small pellet in laboratory settings. However, in our field trials, seedling counts from both HP pellet sizes were regularly different from bare seed in similar ways, with no instances of one size of HP pellet improving late season seedling count over bare seed when the other size did not. Baughman et al. [37] suspected that larger pellets caused more physical resistance to emergence than small pellets during the first 1–2 weeks of their 40-day lab trial, but the slower and longer imbibition, germination, and emergence period in our field trials may have resulted in reduced resistance via increased pellet degradation.

Although the HP pellet shows some promise in terms of protecting seeds from herbicide toxicity, our concurrent, yet-unpublished work has demonstrated that there are many challenges associated with scaling up HP pellet production, handling, and delivery in wildland restoration, which often occurs across hundreds or thousands of hectares. For example, the per-seed weight of HP pellets can be more than 50 times that of bare seed. Alternatively, HP coatings use ca. 75–80% less material than the HP pellet, and are likely to pose fewer transport and delivery challenges. For this reason, the large HP pellet size was dropped from trials in 2020, and two HP coating prototypes for grasses were added. We retained the small HP pellet in the trial as a benchmark for comparison. In that trial, neither HP coating showed improved early or late seedling count over the small HP pellet at any site or for any species. However, only one species (ELEL) at one site (ID) showed lower late season seedling count for HP coatings than small HP pellet, and the vortex coating was associated with leafier seedlings than small HP pellet for both grasses at one site (NV). Therefore, most evidence from the single year trial (2020) with HP coatings suggests that they are as effective as the small HP pellet. Additionally, recent laboratory results using one of the same coatings tested here showed promise for coatings as herbicide protection [47]. Combined with the fact that HP coatings are likely more scalable than the HP pellets because they are easier to produce using conventional seed coating equipment, relative to the HP pellet, and that many refinements are likely possible to the rather rudimentary coatings tested here, we recommend additional testing and refinement of HP coating. The two coating prototypes tested in 2020 were too different in their formulation and production processes to identify specific attributes that should be included in future prototypes, but we suspect that the ability to dissolve quickly when first wetted may be important in restoration applications in arid and semiarid regions.

## Evidence for benefits of herbicide protection seed technologies

The clearest evidence our study can offer of the potential for HP treatments to improve restoration success in conjunction with herbicide use is any improvement in seeding success (seedling count, size) of HP treatments over unprotected bare seed in the presence of herbicide. Similarly, any evidence that the microenvironment provided to the seed from the HP seed treatment is beneficial to seeding outcomes, unrelated to the effects of herbicide, would appear as improved seeding success of HP treatments over bare seed in the absence of herbicide. Finally, evidence of any unintended costs of the HP treatments would appear as any reduction in seeding success for HP treatments compared to bare seed in the presence or absence of herbicide.

Evidence that HP treatments benefited seeding outcomes in the presence of herbicide was present but not ubiquitous in all three years for grasses. Such evidence was present only in the

early season in 2018 for only one species (POSE; ca. 2.5-fold higher seedling count), and predominantly in the early season in 2020 for both grass species (PSSP, ELEL) at two or three of five sites, respectively (ca. 1.7-fold higher count). The strongest evidence came from 2019, where there was consistently 2.7–3 -fold higher seedling count from HP treatments than bare seed for both grass species at all sites in both seasons, whether herbicide was present or not. As previously noted, the large pellet was most commonly the best-performing HP treatment for grasses during the two years in which multiple pellet sizes were tested. In the final year, 2020, HP coatings performed similarly to small HP pellets. Differences in seedling size between HP treatments and bare seed in these comparisons were rare (only ELEL in OR in 2019; where it was 1.9—fold taller for HP pellet). For ARTR, the HP band was the only HP treatment with a higher seedling count relative to bare seed in the presence of herbicide. Evidence of benefits to seedling success from the microenvironment of the HP coatings themselves, in the absence of any effects of herbicide, were occasionally present for grasses and never for ARTR.

Evidence of unintended costs to seedling success associated with HP treatments themselves were less common than evidence of benefits for grasses, and these costs were to seedling size more often than to seedling count. There was no evidence of costs for any species at any site in 2019. ARTR suffered more consistent costs than grasses, with over 70% lower seedling count for HP pellet than bare seed in the only site tested in 2018 (WY) as well as at the site with the most ARTR success in 2020 (OR), and shorter seedlings from carbon strip than bare seed at one site in 2019. Clenet et al. [51] and Baughman et al. [37] also found higher costs for ARTR and suspect it's small seed struggles to emerge from the pellet.

Comparing the benefits to the costs of these HP treatments on seeding success, we did not detect strong and consistent benefits of HP treatments specifically in the presence of herbicide, nor consistent detriments to seeding success. Benefits, when present, were more frequent for grasses than ARTR, and primarily to seedling count, with an average benefit of nearly a two-fold higher seedling count. Costs were less common than benefits for grasses, but more so for ARTR. For grasses, costs were more often to seedling size rather than seedling count, whereas ARTR demonstrated significant costs to seedling count. As observed in laboratory trials [37,51], the tested seed treatments were clearly less suited for use with ARTR than native perennial grasses. Even though our ability to assess technology efficacy was limited due to overall poor seeding conditions experienced across the majority of sites and years, our findings suggest that HP treatments can improve seeding success in some but not all instances. Other multi-species, multi-site field tests of HP technologies [34,36] encountered similar challenges in determining differences in treatment efficacy given low or variable establishment.

The limited efficacy we observed suggests that future prototypes of HP technologies must show more consistent benefits and fewer costs if they are to be a valuable tool for widespread use, at least in dryland restoration. However, we are aware of many promising refinements yet to be tested in field trials, such as different types and rates of activated carbon, different commercial coating techniques, and optimization of coating delivery and dissolvability. Additionally, we stress that future trials must eventually create direct comparisons of HP technologies to the current best practice of combining herbicide with seeding. Our experimental controls involved seeding bare seed into highly invaded plots that did not experience herbicide at any time, which is not a commonly used restoration approach in our region. A more useful control will be bare seed planted into plots that received preemergent herbicide the year prior to seeding, because this is the strategy most likely to be employed by restoration practitioners trying to restore invaded areas [23], and it is the method that the HP approach must outperform for it to be worthwhile for adoption. It is for these reasons that we strongly recommend additional refinement and continued field trials for these technologies before their efficacy is definitively assessed.

## Delivery method was important, litter reduction was not

The effects of seed delivery method in the trials were strong and consistent. Regardless of which seed treatment was applied, surface-sowing into a relatively shallow (2–2.5 cm) furrow was associated with a 2-12-fold higher seedling count than broadcast seeding in nearly all experimental combinations across both years tested. Others have recently shown drill-seeding as a superior method to broadcast seeding for most species in our region [52,53]. Together with our results, it is clear that drilling or shallow furrowing, if possible, is likely to yield higher success than broadcast seeding for grasses. More importantly to our investigations, we found little evidence of any tested HP seed treatments modifying this pattern. In previous work with the large, rectangular HP pellet and other non-carbon pellets and pillows [25,54] it was hypothesized that large, heavy, multi-seed pellets or "pillows" could overcome some disadvantages of broadcast seeding by providing improved seed/soil contact directly (from the pellet itself) and indirectly (from the heavy pellet getting closer to soil surface), but we found no clear support for that hypothesis. The shallow furrows we tested were designed to mimic rows made by commonly used drill-seeding mechanisms, and were much shallower than other furrows under recent investigation (e.g., 15 cm depth investigated by Terry et al. [28]. The major difference between our shallow furrows and those left behind by standard drill-seeding is the lack of seed burial in our method. Therefore, the shallow furrow seeding method described here could likely be applied at scale with existing equipment by simply rerouting the seed delivery tubes of drill seeders to drop seed behind disks rather than between or under disks. Interestingly, our study provides some evidence that ARTR may perform better when seeded unburied in a furrow than when broadcast seeded, though it is often broadcast seeded in practice because of known negative effects of the burial depths achieved by most standard drills [55,56]. Additional laboratory and field trials are needed to explore whether HP seed treatments perform best with or without burial, as well as practical delivery options for prototypes as they continue to be refined.

We found no compelling evidence that reducing litter depth by 60–76% improved herbicide efficacy against invasive annual grasses or forbs in the one year (five sites) in which we tested litter reduction against litter-intact treatments. The only evidence was from the OR site, where an already low cover of IAF (less than 3%) was lowest in reduced litter than intact litter sub-plots in the presence of herbicide. The opposite effect was present in UT, with litter reduction leading to increased IAF cover even in the presence of herbicide. Existing studies on best practice for application of pre-emergent herbicides in this region suggest improved invasive grass control can be attained by selecting sites with less litter, or removing litter via fire or other methods, prior to application [57–60]. Although our litter reduction was achieved by hand raking and only reduced rather than completely removed litter, our findings suggest that pre-emergent herbicide efficacy may be more consistent than previously thought across sites with varying litter presence, at least within the tested range of litter depth and the relatively high herbicide rate we tested. Deeper and more matted litter may still impact efficacy.

## Are our assumptions of the predominant barriers to success accurate?

As previously discussed, we did not observe the kind of consistent seeding failure from bare seed in the presence of pre-emergent herbicides that would be expected if exposure to herbicide was an omnipotent and insurmountable barrier. This suggests that some native seeds can either avoid or tolerate the level of exposure to herbicide we created, at least in certain sites and years. Other research on this topic has had mixed findings, ranging from herbicide appearing to be a strong barrier [61] to additional evidence of some tolerance in certain situations [24,62–64]. Our findings lead us to recommend continued field research into whether

specific delivery methods can improve the success of untreated native seed when used simultaneously with preemergence herbicides. For example, recent research by Terry et al. [28] and Madsen et al. [49] suggest deep furrows (15 cm) created after herbicide application that sweep affected soil to the side can mitigate deleterious herbicide effects on both bare seed and HP technologies. Future trials should continue to explore these methods and, as previously mentioned, compare these single-entry techniques to the more common method of applying herbicide the year before seeding with bare seeds.

Addressing other important barriers to seedling establishment, aside from herbicide exposure, is not the primary focus of HP seed treatments. One of the largest additional barriers in our region is soil and seedbed conditions (moisture, temperature) that are inadequate for seedling survival during the periods after germination but before emergence, which are most commonly in late fall to early spring in our region. We observed patterns of low seedling count despite reasonable rates of cumulative germination, which supports other research highlighting that the life stage between germination and seedling emergence is the most critical bottleneck to seeding success in this region [48,65]. The predominant weather conditions across our 13 trials over three years were below-average precipitation in the winter and spring, and above-average winter temperatures (S2 Fig). Multiple seeding years also included lengthy spring droughts at many sites. Seeding success in our and other related regions is often linked with above-average precipitation and available soil moisture during key periods [66,67], confirming that lack of moisture is a common and major barrier to restoration success that is relatively unavoidable in our region. It is therefore not surprising that we saw low overall establishment across all treatments in most sites and years in our study, and the prevalence of this unaddressed barrier complicated our ability to assess whether—and how significantly—our tested herbicide protection treatments alleviated the barriers of competition and herbicide exposure. We recommend that future trials seeking to provide proof of concept for HP or other seed treatments in arid or semi-arid regions target sites with a reduced threat of inadequate abiotic spring conditions. However, we also stress that efforts to assess the efficacy of seed treatments or any other restoration technique in realistic restoration conditions should anticipate the unavoidable barrier of sporadic establishment conditions by spreading trials across multiple sites and years to mitigate risk of failure due to drought conditions.

## Conclusion

Any approach to consistently establish desired vegetation from seed while simultaneously controlling invasive annual species with a single-entry method would represent a powerful and valuable new paradigm to managing millions of hectares of degraded, semi-arid lands that are increasingly invaded with exotic, annual plants. The desire for such a useful approach has driven nearly a decade of research into herbicide protection seed technology. Our three-year, adaptive field trials demonstrate that some prototypes of this technology can improve seeding outcomes 2-4-fold over untreated bare seed in the presence of herbicide in some but not all situations, but we were generally unable to confirm if the HP seed treatments consistently provided protection of seed from the presumably deleterious effects of pre-emergent herbicide. Prevailing droughty conditions and the inconsistency of herbicide effects on unprotected bare seed surely contributed to our inability to measure consistent effects of the HP seed treatments, but it is nevertheless clear from our results that HP seed treatment efficacy must benefit from additional refinements before it warrants widespread use. Many aspects of herbicide protection technology remain to be refined and tested, from the chemistry of basic ingredients to seed delivery procedures, and there is great potential if a successful combination of specifications is identified. We therefore recommend continued research and development for HP seed

technologies. Additional field trials should continue to utilize multiple sites, species, and years in order to measure efficacy across the large scales at which new solutions to native species restoration are needed.

## Supporting information

**S1 Appendix. Additional methods.**
(DOCX)

**S2 Appendix. Additional results.**
(DOCX)

**S1 Fig. Map of field sites used.** Site names correspond to tables. In Idaho and Wyoming, original sites (1) were relocated to improved sites (2), and trials were never carried out in both sites in the same year within a state.
(TIF)

**S2 Fig. Precipitation and temperature patterns for experimental trials.** Mean temperature anomaly (black needles) and precipitation anomaly (hashed bars) for all tested sites (across top) and planting years (2018, 2019, 2020; top to bottom), for winter (Dec–Feb) and spring (Mar–May) seasons, calculated against 1991–2020 climate normals. Values are percent deviation from the normal precipitation (left axis) and degrees Celsius (right axis). Data are from PRISM Climate Group.
(TIF)

**S3 Fig. Germination timing of bare seed.** Germination of bare seed (placed in shallow seed bags) as a percent of estimated viable seed sown, by species (across top) and planting year (top to bottom). The height of each bar represents the mean cumulative germination of viable seed sown for each site, year, and species, with the portion of this total that occurred within each harvest period indicated by different colors. Asterisks note instances of cumulative germination that exceed 100% of estimated viable seed sown, which suggests field conditions encouraged higher germination rates than petri dish tests used to develop the estimates of viable seed sown. A pre-winter harvest was not made in seeding year 2020, so the pre-spring harvest contains all pre-winter and winter germination for that year.
(TIF)

**S4 Fig. Herbicide effects on bare seed seedling size.** Effect of herbicide on mean seedling height (mm) and mean seedling leaf count of seedlings derived from the bare seed (unprotected) seed treatment for all three years. In 2018 (A), all species and both delivery methods (broadcast, furrow) are pooled. In 2019 and 2020 (B), only furrow delivery data are included. Black stars indicate significant effect (P < 0.05), and all other comparisons are not significant. Error bars are standard errs. Too few seedlings for some species in some years resulted in not enough data to make statistical comparisons.
(TIF)

**S5 Fig. HP and bare seed treatment effects on seedling size.** Differences in seedling size (mean height and leaf count) among seed treatments. In 2018 (A), differences among carbon seed treatments were dependent upon exposure to herbicide (top), but notable differences between carbon treatments and bare seed were not (bottom). Black stars indicate significant difference in ANOVA model (P < 0.05). Bars sharing the same letters within each site for 2020 (B) and for each species in 2019 (C) are not significantly different according to post-hoc Tukey

HSD tests (P < 0.05).
(TIF)

## Acknowledgments

The authors thank Karen Howe, Liz Munn, Elaine York, Bob Unnasch, Brooke and Andrew Gray, Caleb McAdoo, Tom McGinnis, Anne-Marie Raymondi, Scott Walker, and Nathan Long for assistance with study design, coordination, and/or site placement; Michael McCampbell, Berta Youtie, and Matt Benson for assistance with seed acquisition; staff of the EOARC in Burns Oregon for laboratory and research support; Jennifer Ruthruff, Corinna Holfus, Nanda Ramos, Ivy Hinson, Nadav Mouallem, Mary Schneider, Christopher Donovan, Kelsey Flathers, Laynie Saidnawey, Anna Hosford, Emily Ralston, and Isaac Rubinstein for painstaking field and laboratory work; Cameron Duquette and Filiz Erbas for reviewing an earlier version of this manuscript.

## Author Contributions

**Conceptualization:** Owen W. Baughman, Magdalena Eshleman, Jessica Griffen, Chad Boyd, Matthew Cahill, Jay D. Kerby, Corinna Riginos.

**Data curation:** Owen W. Baughman, Magdalena Eshleman.

**Formal analysis:** Owen W. Baughman.

**Funding acquisition:** Chad Boyd, Matthew Cahill, Jay D. Kerby, Corinna Riginos.

**Investigation:** Owen W. Baughman, Magdalena Eshleman, Jessica Griffen, Olga A. Kildisheva, Andrew Olsen, Jay D. Kerby, Corinna Riginos.

**Methodology:** Owen W. Baughman, Magdalena Eshleman, Roxanne Rios, Chad Boyd, Andrew Olsen, Jay D. Kerby, Corinna Riginos.

**Project administration:** Owen W. Baughman, Magdalena Eshleman, Jessica Griffen, Olga A. Kildisheva, Andrew Olsen, Matthew Cahill, Jay D. Kerby, Corinna Riginos.

**Resources:** Owen W. Baughman, Magdalena Eshleman, Jessica Griffen, Roxanne Rios, Chad Boyd, Andrew Olsen, Jay D. Kerby, Corinna Riginos.

**Software:** Owen W. Baughman.

**Supervision:** Owen W. Baughman, Magdalena Eshleman, Jessica Griffen, Roxanne Rios, Olga A. Kildisheva, Andrew Olsen, Jay D. Kerby, Corinna Riginos.

**Validation:** Owen W. Baughman.

**Visualization:** Owen W. Baughman, Corinna Riginos.

**Writing – original draft:** Owen W. Baughman, Magdalena Eshleman, Olga A. Kildisheva, Andrew Olsen, Jay D. Kerby, Corinna Riginos.

**Writing – review & editing:** Owen W. Baughman, Magdalena Eshleman, Jessica Griffen, Roxanne Rios, Chad Boyd, Olga A. Kildisheva, Andrew Olsen, Matthew Cahill, Jay D. Kerby, Corinna Riginos.

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
