## [Decision Letter · Decision Letter 0]

17 Jan 2023

PONE-D-22-34851Assessment of multiple herbicide protection seed treatments for seed-based restoration of native perennial bunchgrasses and sagebrush across multiple sites and yearsPLOS ONE

Dear Dr. Baughman,

Thank you for submitting your manuscript to PLOS ONE. After careful consideration, we feel that it has merit but does not fully meet PLOS ONE’s publication criteria as it currently stands. Therefore, we invite you to submit a revised version of the manuscript that addresses the points raised during the review process.

We look forward to receiving your revised manuscript.

Kind regards,

Ahmet Tansel Serim

Academic Editor

PLOS ONE

3. We note that S2 Fig in your submission contain [map/satellite] images which may be copyrighted. All PLOS content is published under the Creative Commons Attribution License (CC BY 4.0), which means that the manuscript, images, and Supporting Information files will be freely available online, and any third party is permitted to access, download, copy, distribute, and use these materials in any way, even commercially, with proper attribution. For these reasons, we cannot publish previously copyrighted maps or satellite images created using proprietary data, such as Google software (Google Maps, Street View, and Earth). For more information, see our copyright guidelines: http://journals.plos.org/plosone/s/licenses-and-copyright.

a. You may seek permission from the original copyright holder of S2 Fig to publish the content specifically under the CC BY 4.0 license. 

Additional Editor Comments:

Dear authors,

Your manuscript entitled "Assessment of multiple herbicide protection seed treatments for seed-based restoration of native perennial bunchgrasses and sagebrush across multiple sites and years" was reviewed by an academic peer. Please answer all the comments of the reviewer point-by-point in your revised submission.

Sincerely yours,

Editor

Reviewers' comments:

Reviewer's Responses to Questions

**Comments to the Author**

1. Is the manuscript technically sound, and do the data support the conclusions?

Reviewer #1: Yes

2. Has the statistical analysis been performed appropriately and rigorously? 

Reviewer #1: Yes

3. Have the authors made all data underlying the findings in their manuscript fully available?

Reviewer #1: Yes

4. Is the manuscript presented in an intelligible fashion and written in standard English?

Reviewer #1: Yes

5. Review Comments to the Author

Reviewer #1: Detailed explanations were made and all data were shared with all clarity. There are minor revisions and typos in the manuscript. You can see them in the revised manuscript. It is a good experiment, but as you stated, more study is needed to reach clearer results.

6. PLOS authors have the option to publish the peer review history of their article (what does this mean?). If published, this will include your full peer review and any attached files.

Reviewer #1: **Yes: **FİLİZ ERBAS

---

## [Author Response · Author response to Decision Letter 0]

10 Feb 2023

Changes to the manuscript in response to in-line edits by the reviewer:1) All instances of AC (activated carbon) have been replaced with HP (herbicide protection) to be consistent, because it is clear that our HP treatments rely entirely on AC as the active ingredient. 2) Misspelling in Table 2 (surface) corrected (surface). 3) The typo “vibility” in table S1 was corrected, thank you. 4) Instances of AC reference in Table 3 and footnotes were changed to HP to maintain consistency with text. 5)Abbreviation of “Kam” in Table 3 changed to “Kamterter” to be consistent with text. 6)Typo of “acre” in Table 3 footnotes corrected to hectare (“ha”). 7) Typo of species (ARTRWY) corrected in Table 3 footnotes (ARTR). 8) Space added to “seed lot” in Table 3 footnote. 9) Instance of repeated phrase at line 500 was reworded. 10) Space added at line 637 to correct typo. 11) Reviewer comment at line 501: “You say “improvement in seeding success” above…” We will keep the current wording (“reduction in seeding success”) because if the seed technologies perform more poorly than bare seed, that is not an ‘improvement’ in success of bare seed over the seed technologies, because the bare seed is the baseline for comparison. Therefore, referring to it as a reduction in success for the seed technologies is an accurate way to refer to this difference. We are happy to reconsider if we have misinterpreted your comment. 

Editor comments: 1) "Please ensure that your manuscript meets PLOS ONE's style requirements, including those for file naming." We reviewed the Style Guide and corrected the names of supporting information files and corrected a few cels in table to remove “returns” used for spacing. We believe the rest of our submission adheres to the style guide but will be happy to make additional corrections if we have missed something. 2) "In your Methods section, please provide additional information regarding the permits you obtained for the work. Please ensure you have included the full name of the authority that approved the field site access and, if no permits were required, a brief statement explaining why." Thank you for pointing out this omission. We have added wording (line 129-130) explaining the permits obtained for the three sites that needed it, and explaining that the remainder of the sites did not require formal permitting. 3) "We note that S2 Fig in your submission contain [map/satellite] images which may be copyrighted." Thank you for spotting this, our apologies for this oversight. We have reworked S2 fig to contain public domain imagery from the USGS link provided. 4) "Please review your reference list to ensure that it is complete and correct. If you have cited papers that have been retracted, please include the rationale for doing so in the manuscript text, or remove these references and replace them with relevant current references." We have double checked our references and compared them against Retraction Watch’s list of retractions. We found several systemic format errors that we corrected (space after “doi:”, “from” after “Available”), corrected several typos, and found no instances of references to retracted works. We found no other issues with the references and in-text citations, but will be happy to address any that we have missed.

---

## [Decision Letter · Decision Letter 1]

14 Mar 2023

Assessment of multiple herbicide protection seed treatments for seed-based restoration of native perennial bunchgrasses and sagebrush across multiple sites and years

PONE-D-22-34851R1

Dear Dr. Baughman

We’re pleased to inform you that your manuscript has been judged scientifically suitable for publication and will be formally accepted for publication once it meets all outstanding technical requirements.

Kind regards,

Tunira Bhadauria, Ph.D.

Academic Editor

PLOS ONE

Reviewers' comments:

Reviewer's Responses to Questions

**Comments to the Author**

1. If the authors have adequately addressed your comments raised in a previous round of review and you feel that this manuscript is now acceptable for publication, you may indicate that here to bypass the “Comments to the Author” section, enter your conflict of interest statement in the “Confidential to Editor” section, and submit your "Accept" recommendation.

Reviewer #1: All comments have been addressed

Reviewer #2: All comments have been addressed

2. Is the manuscript technically sound, and do the data support the conclusions?

Reviewer #1: Yes

Reviewer #2: Yes

3. Has the statistical analysis been performed appropriately and rigorously? 

Reviewer #1: Yes

Reviewer #2: Yes

4. Have the authors made all data underlying the findings in their manuscript fully available?

Reviewer #1: Yes

Reviewer #2: Yes

5. Is the manuscript presented in an intelligible fashion and written in standard English?

Reviewer #1: Yes

Reviewer #2: Yes

6. Review Comments to the Author

Reviewer #1: All the comments have been addressed. Manuscript is technically sound, presented in an intelligiblle fashion and written in standard English. And the data support the conclusion. I wish you succes in your further studies.

Reviewer #2: The work is focused upon multiple herbicide protection seed treatments for seed-based

restoration of native perennial bunchgrasses and sagebrush across multiple sites and

years analysis.

1. Use °C in place of C for temperature.

2. Clarify little bit more about pre-emergent herbicides.

3. Mention the reason why authors used multiple sites and

years analysis.

4. Conclusion should be more precise and summarized.

7. PLOS authors have the option to publish the peer review history of their article (what does this mean?). If published, this will include your full peer review and any attached files.

Reviewer #1: **Yes: **FİLİZ ERBAŞ

Reviewer #2: **Yes: **Dr. Bijendra Kumar Singh

<quillbot-extension-portal></quillbot-extension-portal>

---

## [Editor Report · Acceptance letter]

20 Mar 2023

PONE-D-22-34851R1 

Assessment of multiple herbicide protection seed treatments for seed-based restoration of native perennial bunchgrasses and sagebrush across multiple sites and years 

Dear Dr. Baughman:

I'm pleased to inform you that your manuscript has been deemed suitable for publication in PLOS ONE. Congratulations! Your manuscript is now with our production department. 

Kind regards, 

on behalf of

Dr. Tunira Bhadauria 

Academic Editor

PLOS ONE